# An Evaluation of the Water Quality Characteristics of Shipboard Sewage Disposal and Usability Based on Water Quality Enhancement

**Young-Ik Choi [1], Hyeon-Jo Ji [1], Dae-Yeol Shin [1], Sana Mansoor [1], Min-Ji Kwan [1], Seung-Chul Lee [1], Jin-Hee Jung [1], Byung-Gil Jung [2], Nak-Chang Sung [1] and Jei-Pil Wang [3],***

[1]   Department of Environmental Engineering, Dong-A University, Busan 49315, Korea;
      youngik@dau.ac.kr (Y.-I.C.); whwh93@hanmail.net (H.-J.J.); dys1023@hanmail.net (D.-Y.S.);
      sanamansoorahmad@gamil.net (S.M.); kwonminji56@gmail.net (M.-J.K.); ui0h@hanmail.net (S.-C.L.);
      jjh8014@dau.ac.kr (J.-H.J.); ncsung@dau.ac.kr (N.-C.S.)
[2]   Department of Environmental Engineering, Dong-Eui University, Busan 47340, Korea; bgjung@deu.ac.kr
[3]   Department of Metallurgical Engineering, Pukyong National University, Busan 48547, Korea
*    Correspondence: jpwang@pknu.ac.kr; Tel.: +82-5162-9634-1

**Abstract:** International Maritime Organization recognizes vessel-induced pollution as a global issue. The designation of the Baltic Sea as the special uncontaminated area was the beginning of the regulations for preventing marine pollution. In this regard, a process is needed which meets the provisions of MEPC. 227(64) for the specificity and constrained conditions of vessels, removes both nitrogen and phosphorus, requires convenient operation and low construction cost and is little affected by the load variation of inflow. This study used an SBR(sequencing batch reactor) and MBR(membrane bioreactor) combined process to iterate the stirring and aeration process and maintained the ratio of raw wastewater parameters (C:N:P) to be 10:3:1 in order to assess the quality and future availability of ultimate outflow in each time period of the stirring and aeration process. The removal efficiencies of COD and T-N exceeded 90% and 93% respectively. However, a detailed mechanism will be identified by a further study on nitrogen removal issues like DO aeration condition, stirring duration, ORP and $NO_3$. As the removal efficiency of T-P exceeded 95%, the SBR and MBR process formed anaerobic and aerobic conditions without a separate coagulation process for removing phosphorus, thereby enabling easy phosphorus release and uptake. The optimal stirring and aeration condition seems to be 70–50 min. A further study will be efficiently conducted by focusing on the water quality criteria of the Maritime Environment Protect Committee. 227 (64) for *E. coli* and chlorine and a detailed mechanism.

**Keywords:** IMO; shipboard sewage treatment plant; SBR; MBR; nutrient

## 1. Introduction

Korea is a leading country in the marine shipbuilding industry based on its large-scale facilities. It records a deficit trade balance of 13.3 billion dollars and surplus of 37.8 billion dollars, performing the role of a driving force in national economic growth [1]. An increase in living standards requires the necessity of marine tourism [2].

In marine tourism, the cruise industry has rapidly grown in recent years, gradually transforming into a tourism product for the general public with the increase in distributors [3]. The most significant characteristic of a cruise is that it utilizes a luxury cruise liner equipped with the luxury services of high-class hotels (luxury rooms, gourmet food, swimming pools, banquet halls, restaurants) [4]. The World Tourism Organization also predicts that the cruise tourism industry will increase until 2020,

and the government is putting forth policies on fostering a high-class marine tourism culture of the future [5].

The US has the most active cruise industry, followed by the European and Asian markets. When comparing port entry numbers for proportions in the cruise industry, the Caribbean region of the US was the highest at 34.4%, followed by the Mediterranean at 21.7% making it a major key cruise tourism region [6]. However, cruise ships are exacerbating various pollutants due to an increase in ship traffic, which has resulted in a rising trend in instances of ecological devastation and damage [7].

International Maritime Organization (IMO) recognizes environmental pollution due to ships as a global issue, and regulations for pollution control by wastewater from ships that were adopted in 1973 by the Marine Environment Protection Committee (MEPC) were ratified in 2002 in Norway. In September of 2003, the Wastewater Prevention Agreement MARPOR annexes 73/78 came into effect [8,9].

The MEPC. 200(62) designates the Baltic seas as a special uncontaminated area, and is a control standard for marine pollution prevention. In 2012 the MEPC. 227(64) was adopted and replaced the original MEPC. 159(55). Additionally, it only regulates the items of colon bacillus, TSS, BOD5, COD and pH, but does not include control standards based on the occurrence of problems such as eutrophication [10].

MEPC. 227(64) presented removal standards of T-N and T-P, and the detailed contents of T-N stipulate a removal rate of 20 Qi/Qe mg/L or 70%, and T-P stipulates a removal rate of 1.0 Qi/Qe mg/L or 80%, but this was not a settled matter due to issues in technical skills at the time of adoption. However, the majority of the countries involved determined at the 67th conference that they would be able to satisfy the MEPC standards [11,12]. Based on the addition of reinforced water quality standards in human waste disposal for ships in the MEPC. 227(64), development is being carried out on environmentally friendly technologies such as dual-structured hull technology, non-toxic paint development, exhaust reduction and ballast water treatment technology [1,12].

## 2. Materials and Methods

### 2.1. Experimental Materials

For the raw wastewater used in this study, regular sewage was used to manufacture experimental artificial wastewater. A fixed C:N:P ratio with maximum proportionality was manufactured for the raw wastewater, and a fixed hourly flow rate was made to flow into the device through a standard capacity influent pump. Table 1 depicts the average properties of the raw wastewater.

**Table 1.** Characteristics of the artificial wastewater.

| Parameters | | Units | Measured Values |
|---|---|---|---|
| | | | Avg. |
| Influent and Effluent flow rate | | L/hr | 2 |
| Temperature | | °C | 25 |
| pH | | - | 7.2 |
| Condition | $COD_{Cr}$ | mg/L | 300 |
| | T-N | mg/L | 150 |
| | T-P | mg/L | 90 |

### 2.2. Experiment Apparatus and Methodology

#### 2.2.1. Experiment Apparatus

The overall standard of this study's experiment apparatus was W 300 mm × L 567 mm × H 502 mm, and the flow rate of raw wastewater was 2 L/h for each condition.

The devices used in this study consisted of a down flow anaerobic reactor, screen, bioreactor, and membrane reactor, and in the case of raw-water, considering the water runoff, a constant water level was maintained through a pump.

In the case of raw-wastewater, it was induced to flow into the down flow anaerobic reactor, and by installing the screen of mesh material in the anaerobic reactor, the SS was filtered first and the filtered wastes were piled up at the bottom of device, thereby inducing the formation of an anaerobic tank.

In the bioreactor, while the reaction is carried out in the SBR process as a SBR and MBR merging process, the aeration time in the MBR separator (using a hollow fiber membrane) was 3 min stopped/7 min operating and producing produced water through a pump. Figure 1 shows a front view of the shipboard advanced wastewater treatment at a bench-scale.

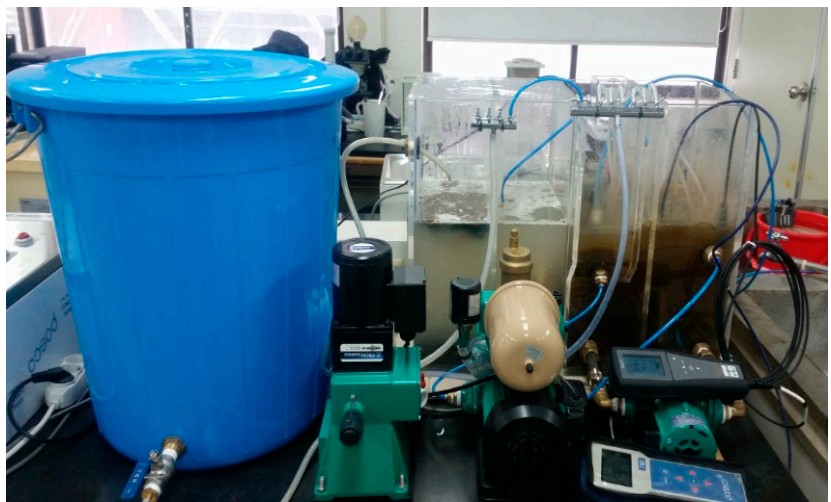

**Figure 1.** Front view of the shipboard advanced wastewater treatment at a bench-scale.

Figure 2 shows a picture of effluent water. A diagram of the process and picture of the apparatus is shown in Figure 3.

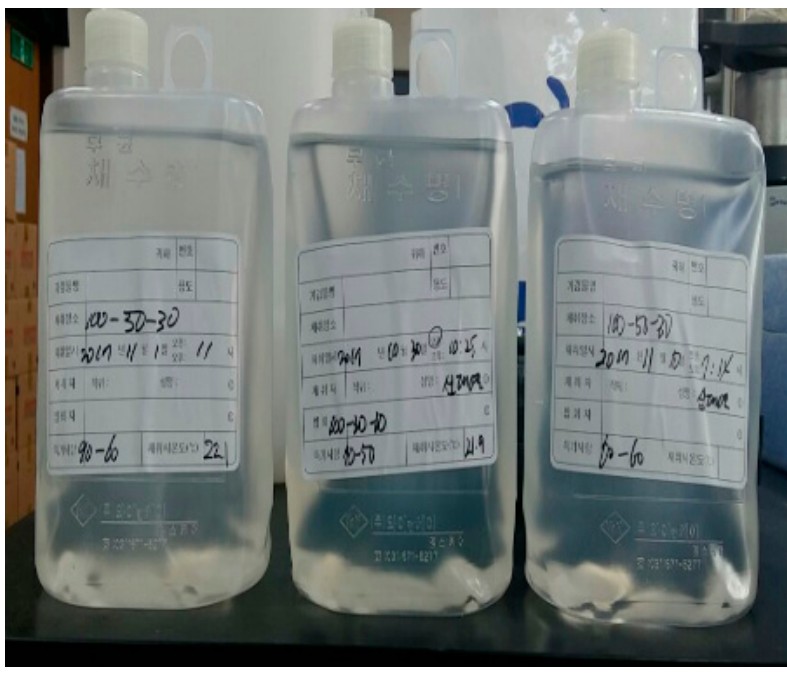

**Figure 2.** A picture of effluent water.

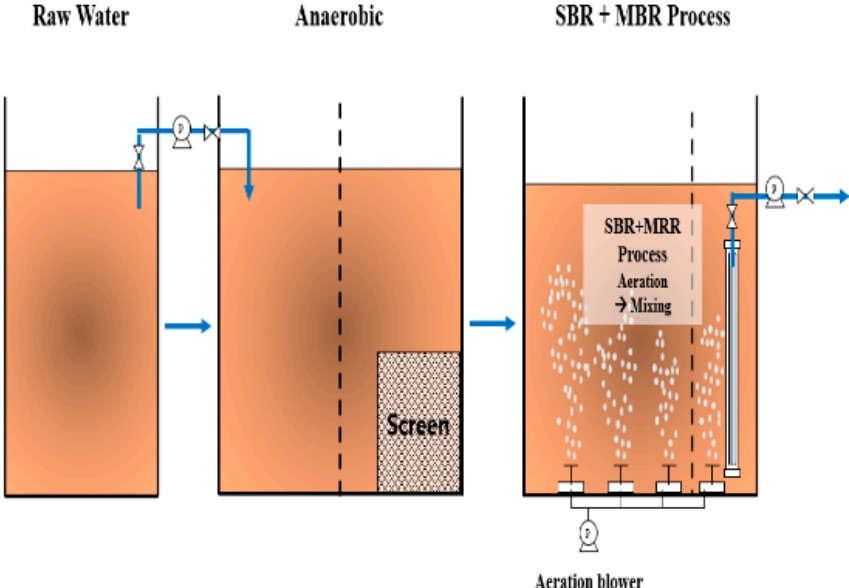

**Figure 3.** Schematic diagram of the Bench-scale shipboard STP.

### 2.2.2. Experimental Method

The existing SBR method is a process carried out through aeration → precipitation → mixing → precipitation, but because the SBR and MBR compound process has the advantage that it can reduce precipitation time, this study carried out a test through the repetitive operation of a mixing → aeration process. The C:N:P ratio of raw wastewater properties was fixed at 10:3:1, and the time required for setup time for mixing: aeration was changed to analyze the removal efficiency of the concentration of the T-N, T-P and $COD_{Cr}$ of the final effluent. Table 2 shows the detailed operating conditions of the reactor.

**Table 2.** The detailed operating conditions of the reactor.

| Parameters | | Units | Conditions | | | |
|---|---|---|---|---|---|---|
| Anaerobic phase | Drain flow | L/h | Auto | | | |
| | Anaerobic phase | | O | | | |
| | MLSS | mg/L | 2000 | | | |
| SBR reactor | Aeration | min | 60 | 50 | 40 | 60 |
| | DO | mg/L | | 3 | | |
| | Mixing | min | 60 | 70 | 80 | 90 |
| | DO | mg/L | | 0.2 | | |
| MBR (aeration period) | Drain flow | L/h | 2 | | | |
| | Drain (On/Off) | min | 3/7 | | | |
| HRT | | min | 360 | | | |

## 3. Results and Considerations

### 3.1. CODCr Concentration Changes and Removal Efficiency Based on Mixing:Aeration Conditions

The C:N:P ratio of raw water properties was fixed at 10:3:1, and the C/N proportion ratio was maintained above 2.5 for the smooth removal of nitrogen. Then, changes in $COD_{Cr}$ concentration and removal efficiency based on changes in the setup time of mixing: aeration were analyzed.

The $COD_{Cr}$ concentration of the raw water, which is artificial wastewater, was 300 mg/L, and treated water concentration of mixing: aeration conditions (60–60 min, 70–50 min, 80–40 min and 90–60 min) were each 29 mg/L, 16 mg/L, 50 mg/L and 29 mg/L, and was the lowest at a mixing:

aeration of 70–50 min and 90–60 min. The removal efficiency for each condition of mixing: aeration (60–60 min, 70–50 min, 80–40 min and 90–60 min) were each 90.33%, 94.67%, 83.33% and 94.67%. Typically, the removal of organic matter takes place in the aeration stage, and is known to oxidize organic matter by microorganisms that use oxygen. This study did not conduct a separate analysis of BOD, but given that the removal efficiency of $COD_{Cr}$ exceeded 85%, BOD was also likely removed. Also, removal efficiency was at its lowest when mixing: aeration was 80–40, and thus aeration time should also be taken into consideration during $COD_{Cr}$ removal. Figure 4 shows raw water and treated water changes based on changes in mixing: aeration time. Figure 5 shows the changes in removal efficiency based on changes in mixing: aeration times.

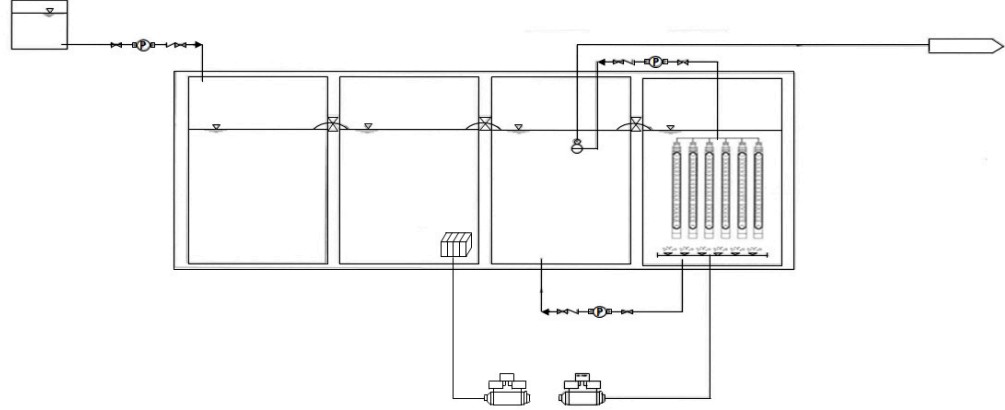

**Figure 4.** Schematic figure of the bench-scale shipboard STP.

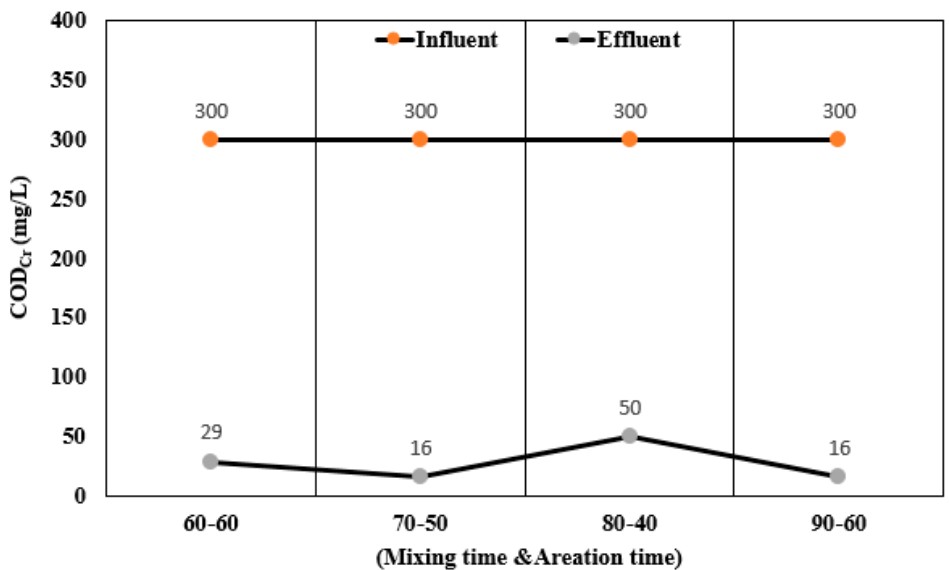

**Figure 5.** Raw water and treated water changes based on changes in mixing: aeration times ($COD_{Cr}$).

### 3.2. Changes in T-N Concentration and Removal Efficiency Based on Mixing:Aeration Conditions

This study analyzed changes in T-N concentration and removal efficiency based on changes in the setup time of mixing: aeration.

The raw water T-N concentration, which is an artificial wastewater, was 150 mg/L. The treated water concentration of each mixing: aeration condition (60–60 min, 70–50 min, 80–40 min and 90-60 min) was 4.512 mg/L, 6.432 mg/L, 7.968 mg/L and 3.984 mg/L each, and mixing: aeration was lowest at 90-60 min. The removal efficiency of each mixing: aeration condition (60–60 min, 70–50 min, 80–40 min and 90–60 min) was 96.99%, 95.71%, 94.69% and 97.34% each.

Based on the above results, the removal efficiency of T-N was positive and the formation of anaerobic conditions took place. However, additional research is needed on DO, ORP, and $NO_3$ to make a more accurate determination on detailed mechanisms.

Figure 6 shows raw water and treated water changes based on changes in mixing: aeration time. Figure 7 shows changes in removal efficiency based on changes in mixing: aeration times.

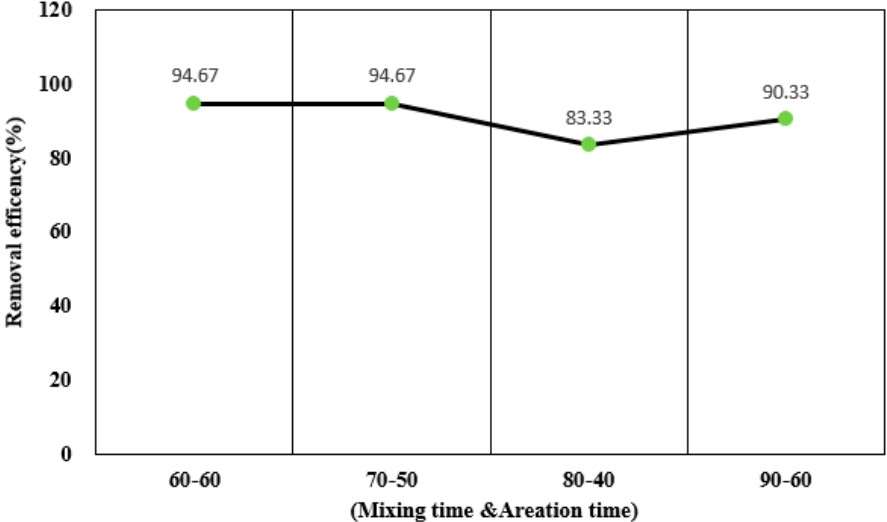

**Figure 6.** Changes in removal efficiency based on changes in mixing: aeration times ($COD_{Cr}$).

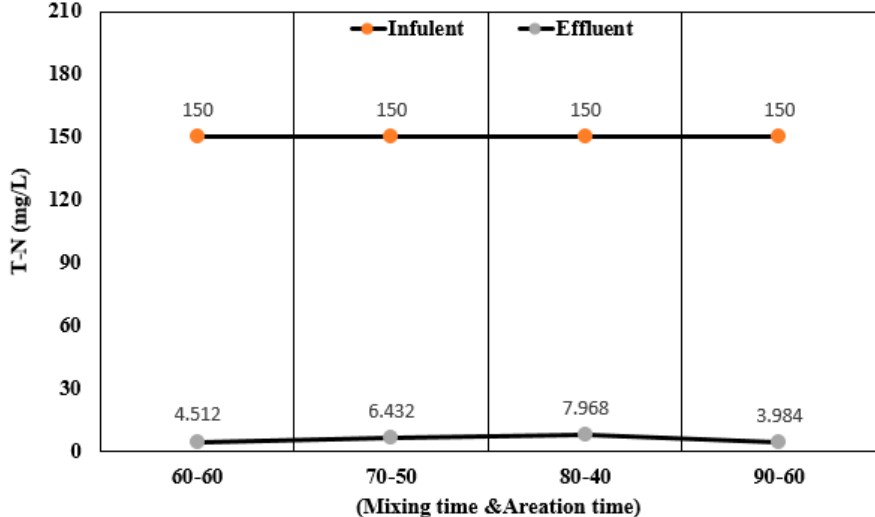

**Figure 7.** Raw water and treated water changes based on changes in mixing: aeration times (T-N).

*3.3. Changes in T-P Concentration and Removal Efficiency Based on Mixing:Aeration Conditions*

This study analyzed changes in T-P concentration and removal efficiency based on changes in the setup time of mixing: aeration.

The raw water T-P concentration, which is an artificial wastewater, was 150 mg/L. The treated water concentration for each mixing: aeration condition (60–60 min, 70–50 min, 80–40 min and 90–60 min) was 3.504 mg/L, 3.36 mg/L, 3.552 mg/L, and 3.504 mg/L each, and lowest at an mixing: aeration of 70–50 min. The removal efficiency for each mixing: aeration condition (60–60 min, 70–50 min, 80–40 min and 90–60 min) was 96.11%, 96.27%, 96.05% and 96.11% each. Based on the above results, the anaerobic and aerobic conditions were formed and phosphorus release and intake went smoothly.

Figure 8 shows raw water and treated water changes based on changes in mixing: aeration time. Figure 9 shows changes in removal efficiency based on changes in mixing: aeration time.

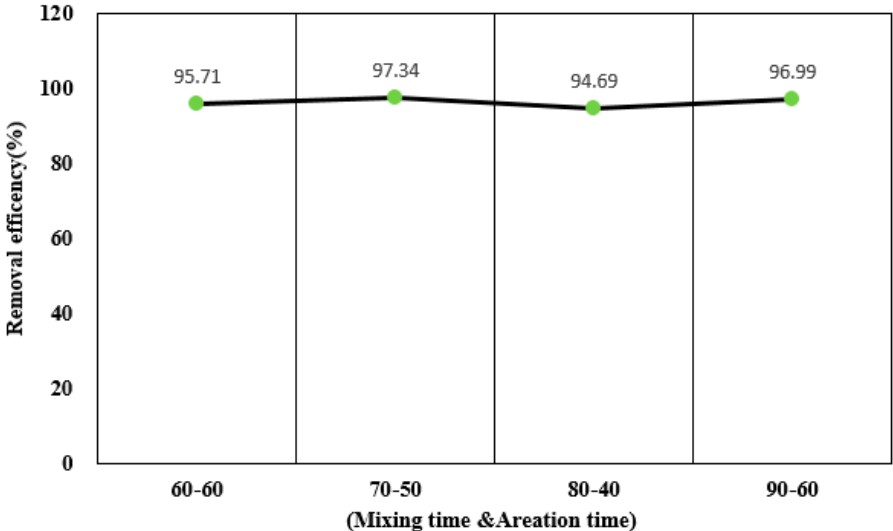

**Figure 8.** Changes in removal efficiency based on changes in mixing: aeration times (T-N).

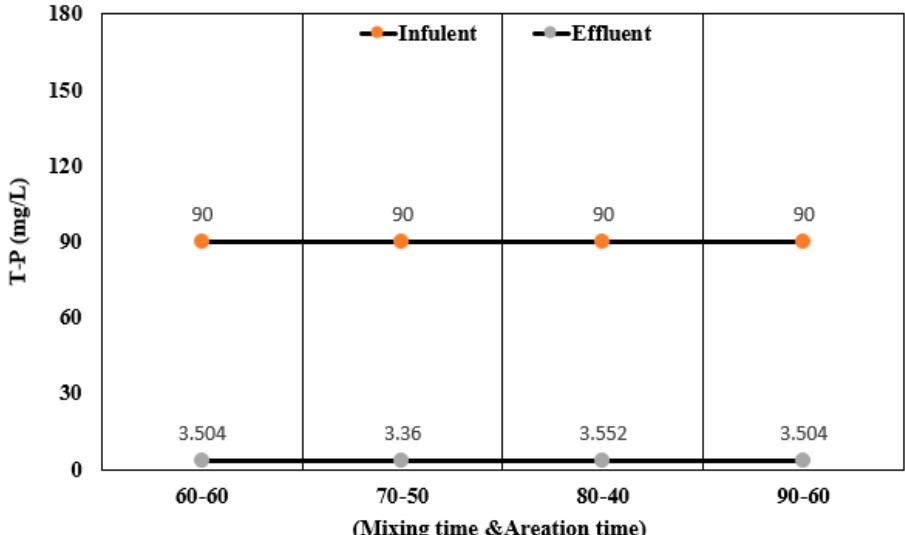

**Figure 9.** Raw water and treated water changes based on changes in mixing: aeration times (T-P).

*3.4. Maximum Efficiency for Respective Condition and MEPC. 227(64) Regulation Compliance*

At the 64th session of the Marine Environment Protection Committee (MEPC), International Maritime Organization (IMO), MEPC. 227(64) was adopted, replacing MEPC. 159(55). Consequently, the regulations were strengthened for nitrogen (less than or equal to 20 $Q_i/Q_e$ mg/L or 70% removal) and phosphorus (less than or equal to 1.0 $Q_i/Q_e$ mg/L or 80% removal) in advanced shipboard wastewater treatment. In this study, the results according to the optimal conditions and compliance with MEPC. 227(64) were confirmed. Table 3 shows the MEPC. 227(64) and research result comparison.

**Table 3.** MEPC 227(64) and Research Result Comparison.

| Remark | MEPC. 227(64) | Research Result (Maximum Efficiency) |
|---|---|---|
| Chemical Oxygen Demand (COD) | Less than or equal to 125 mg/L | 16 mg/L Removal rate: 94.67% |
| Total Nitrogen (T-N) | Less than or equal to 20 Qi/Qe mg/L or 70% removal | 3.984 mg/L Removal rate: 97.34% |
| Total Phosphorus (T-P) | Less than or equal to 1.0 Qi/Qe mg/L or 80% removal | 3.36 mg/L Removal rate: 96.27% |

The COD concentration suggested in MEPC. 227(64) is less than or equal to 125 mg/L on average, and in this study, it is 16 mg/L, showing a satisfying numerical value. In the case of T-N, the concentration suggested by MEPC. 227(64) is less than or equal to 20 mg/L or at least 70% removal for raw-water and influent water, but in this study, the results are 3.984 mg/L and 97.34% removal, satisfying the regulation. Furthermore, in the case of T-P, the concentration suggested in MEPC. 227(64) is 1.0 or at least 80% removal rate for raw-water and influent water. In this study, the T-P concentration is 3.36 mg/L and the removal rate is 96.27%, i.e., the removal rate is satisfied but the concentration is unsatisfactory. Nevertheless, it is determined that there will be no problem since the removal rate for the raw-water satisfies the MEPC. 227(64) regulation. It is due to the higher concentration of T-P compared to the concentration of raw-water, and in future, specific mechanism and other studies will be necessary for phosphorus treatment such as condensation for its removal.

## 4. Conclusions

This study developed a compact sewage treatment plant by applying both SBR and MBR processes and evaluated the performance of the device in terms of disposal, water quality and availability. The SBR process is suitable for small-scale sewage treatment, can treat a large quantity of waste water in a smaller site and has many other advantages, as compared to the conventional activated sludge process. On the other hand, the MBR process needs no sedimentation basin and shows high treatment efficiency. The findings of this study are summarized as follows.

(1) The $COD_{Cr}$ removal efficiencies were 90.33%, 94.67%, 83.33% and 94.67%. The average efficiency of over 90% indicated positive performance in treating organic matters.

(2) The T-N removal efficiencies were 96.99%, 95.71%, 94.69% and 97.34%, of which the average exceeded 93%. However, further details of the mechanism need to be identified by investigating nitrogen removal issues such as DO aeration condition, stirring duration, ORP and $NO_3$.

(3) The T-P removal efficiencies were 96.11%, 96.27%, 96.05% and 96.11%, of which the average exceeded 95%. The SBR and MBR process formed anaerobic and aerobic conditions without a separate coagulation process for removing phosphorus, thereby enabling easy phosphorus release and uptake.

(4) As a result of comparison with MEPC. 227(64) regulation, the treated water's T-P concentration is unsatisfactory, but since the removal rate is satisfied, it will not be a problem, and in future, an additional study will be necessary for T-P removal.

From the above results, it turned out that 70–50 min was the optimal condition achieving the highest removal efficiency. If a further study clarifies the water quality criteria of MEPC. 227(64) for *E. coli* and chlorine and the details of mechanism, the proposed vessel sewage treatment plant will be more efficient

**Author Contributions:** Data curation, B.-G.J., M.-J.K. and J.-H.J.; Formal analysis, D.-Y.S.; Investigation, H.-J.J.; Project administration, Y.-I.C.; Resources, S.M.; Validation, N.-C.S.; Visualization, S.-C.L.; Writing—review & editing, J.-P.W.

**Acknowledgments:** This study was conducted in 2017 under the support of the National Research Foundation of Korea financed by the government (Ministry of Science, ICT and Future Planning) (No. 2017R1A2B4011847).

**Conflicts of Interest:** The authors declare no conflicts of interest.

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
