# Peer review of "An Evaluation of the Water Quality Characteristics of Shipboard Sewage Disposal and Usability Based on Water Quality Enhancement"

_applsci, doi:10.3390/app9030418_

Reviewer 1 Report

The technological problem is very important. But your paper needs many improvements: 

1. Introduction section is too long. Many information contained in lines 32 -74 are interesting , but not important from the point of view of the topic of the paper. It must be shortened. 

2. You deal in your research with wastewater, not with water. So please replace term "water" by "wastewater" one.  There are many mistakes like this in your article (ncluding "raw water" instead "raw wastewater".

3. Section 2.1 Experimental materials: I did not find in Introduction section data concerning the quality of cruise ships sewage. Is this wastewater characteristic special comparing to domestic wastewater?  I understand that the concentration values in Table 1 present average parameters of  wastewater coming from cruise liner ? How did you find them?

4. Section 2.2.2 lines 118-119 Are you sure that "precipitation" is a proper term? 

5. Table 2 I did not find any explanation why you applied in your research the values of parameters contained in Table 2. 

6. How did you handle with hollow fiber membrane clogging? 

7.  You are proposing the "compact device" for wastewater treatment. Do you have any idea how to handle with sludges? 

8. Have you examined  anaerobic reactor effluent quality?   Or SBR effluent?  You present  the efficiency of compact device and we do not know how work the elements of this technological scheme.  

Author Response

Dear Editors

Thank you for your kind direction to improve my article and I revised it as you requested.

Requests

Comments

1. Introduction section is too long. Many information contained in lines 32 - 74 are interesting , but not important from the point of view of the topic of the paper. It must be shortened.

We summarize the introduction.

2. You deal in your research with wastewater, not with water. So please replace term "water" by "wastewater" one. There are many mistakes like this in your article (including "raw water" instead "raw wastewater".

from water to wastewater

3. Section 2.1 Experimental materials: I did not find in Introduction section data concerning the quality of cruise ships sewage. Is this wastewater characteristic special comparing to domestic wastewater?  I understand that the concentration values in Table 1 present average parameters of  wastewater coming from cruise liner ? How did you find them?

It is a domestic wastewater. This research used ratio of Korea wastewater treatment plant raw wastewater.

4. Section 2.2.2 lines 118-119 Are you sure that "precipitation" is a proper term?

yes, right word

5. Table 2 I did not find any explanation why you applied in your research the values of parameters contained in Table 2.

Those are explained at 2.2.1 and 2.2.2

6. How did you handle with hollow fiber membrane clogging?

We maintained the cleanliness and aeration pressure of hollow fiber membrane to prevent from clogging.

7.  You are proposing the "compact device" for wastewater treatment. Do you have any idea how to handle with sludges?

We are proposing compact device it deals with miniaturization of device that helpful to manage ship wastewater sludges.

8. Have you examined  anaerobic reactor effluent quality? Or SBR effluent? You present  the efficiency of compact device and we do not know how work the elements of this technological scheme.  

we are not checking that but we will checking an anaerobic reactor effluent quality and SBR effluent.

Reviewer 2 Report

all abbreviations should be clarified in the whole manuscript

all abbreviations should not be used in Abstract

a schematic figure of both systems (SBR and MBR) would be helpful

Tables are too big and should be improve; the column with units should be reject and units should be add to left or to right column

lines 105-107 are totally unclear

explain "mixing:aeration conditions"

for minutes - "mins" ar "min"?

Figure captions are completely incomprehensible and ambiguous

Conclusions - these values (in %) are different repetitions? 

Author Response

Dear Editor

Happy new year and you are good luck always.

Thank you for your kind direction that help imporove my article.

I revised the article uploaded and rewrote the article.

Thank you again.

1 all abbreviations should be clarified in the whole manuscript

check it

2. all abbreviations should not be used in Abstract

check it

3. a schematic figure of both systems (SBR and MBR) would be helpful

add a schematic figure

4. Tables are too big and should be improve; the column with units should be reject and units should be add to left or to right column

check it

5. lines 105-107 are totally unclear

check it

6. explain "mixing:aeration conditions"

explain at 2.2.2

7. for minutes - "mins" ar "min"?

all “min“ change

8. Figure captions are completely incomprehensible and ambiguous

check it

9. Conclusions - these values (in %) are different repetitions?

check it

Round  2

Reviewer 1 Report

I wrote You, that You were dealing in your research with wastewater, not with water. So please replace term "water" by "wastewater" one. There are many mistakes like this in your article (including "raw water" instead "raw wastewater".

There still many places where you use "water" instead "wastewater - lines: 114, 124, 128 (Fig. 5), 134, 142, 151, 152, 157, 160 (Fig. 9), 160, 183, 187, 192, 206. I am not sure, that I found them all.  I WORD you can apply "the replace" option.